# How deep is your art: An experimental study on the limits of artistic understanding in a single-task, single-modality neural network

**Mahan Agha Zahedi** [ID][1][☯]*, **Niloofar Gholamrezaei**[2☯¤], **Alex Doboli** [ID][1☯]

**1** Department of Electrical and Computer Engineering, Stony Brook University, Stony Brook, New York, United States of America, **2** School of Art, Texas Tech University, Lubbock, Texas, United States of America

☯ These authors contributed equally to this work.
¤ Current address: Department of Humanities, Regis College, Weston, Massachusetts, United States of America
* mahan.aghazahedi@stonybrook.edu

**Data Availability Statement:** The code implementation and the information to reconstruct the datasets are open-accessed via the following

## Abstract

Computational modeling of artwork meaning is complex and difficult. This is because art interpretation is multidimensional and highly subjective. This paper experimentally investigated the degree to which a state-of-the-art Deep Convolutional Neural Network (DCNN), a popular Machine Learning approach, can correctly distinguish modern conceptual art work into the galleries devised by art curators. Two hypotheses were proposed to state that the DCNN model uses Exhibited Properties for classification, like shape and color, but not Non-Exhibited Properties, such as historical context and artist intention. The two hypotheses were experimentally validated using a methodology designed for this purpose. VGG-11 DCNN pre-trained on ImageNet dataset and discriminatively fine-tuned was trained on handcrafted datasets designed from real-world conceptual photography galleries. Experimental results supported the two hypotheses showing that the DCNN model ignores Non-Exhibited Properties and uses only Exhibited Properties for artwork classification. This work points to current DCNN limitations, which should be addressed by future DNN models.

## Introduction

While the study of art has traditionally been the focus of art history, aesthetics, philosophy, psychology and other related areas, advances in Artificial Intelligence (AI) and Machine Learning (ML) have enabled new avenues of inquiry, like devising novel computational models, such as Deep Neural Networks (DNNs), to automatically classify, recognize, and generate artwork [1]. It has been reported that DNNs can identify art genres, artists, and the time range of an art object's creation [1–6]. Applications of these DNN models include helping art curators and historians understand, explore, and navigate through the numerous artworks in museums, galleries, and online sources. Investigating AI/ML models also offers insight on how low-level visual features can lead towards the discovery of high-level semantic knowledge, like image content and object significance, and thus possibly result in unsupervised knowledge discovery, including tacit knowledge, abstractions, and conceptual reasoning.

link on GitHub. https://github.com/aghazahedim/
How-Deep-is-Your-Art/tree/main.

**Funding:** The author(s) received no specific
funding for this work.

**Competing interests:** The authors have declared
that no competing interests exist.

Any attempt to mechanically analyze artwork should reflect the nature of art and how it differs from other types of images. Jerrold Levinson's philosophy of art offers a concrete definition of art, inclusive of both traditional and conceptual works of art. Levinson considers a work of art to incorporate two major properties, Exhibited Properties (EXPs) and Non-Exhibited Properties (NEXPs) [7]. EXPs are the visible elements of art objects, such as color, texture, and form. NEXPs are the ones that are essential artistic aspects of artwork, though they are not simply visible in art objects. NEXPs are accessible by relating art objects and EXPs to human history, culture, and individuals who created the work [8]. Fig 1 summarizes the two kinds of properties.

EXPs sometimes directly point to NEXPs but other times they do not. Rather, understanding NEXPs may require complex contextualization and interpretation. Moreover, some artwork contains more EXPs than NEXPs, while some work, particularly artwork identified as conceptual art, is highly loaded with NEXPs. For example, the painting "The Accident" by William Geets (1899) (Fig 1(c)-bottom) is a narrative figurative work that can be understood to a great extent just by looking at the picture, as it contains more EXPs than NEXPs. In contrast, the famous work "Fountain" by Marcel Duchamp (1917) is meaningful mainly based on its NEXPs (Fig 1(c)-top).

Based on Levinson's theory, what makes Duchamp's urinal art, and hence different from other mass-produced urinals, is not its shape, color, or style but the intention of the artist toward the object in relation to the historical discourse of art [9].

The DNN models used for computational art-related activities rely on EXP processing. While EXPs might be sufficient to tackle some art genres, like iconoclasm and medieval European religious art [1], it is unclear if EXPs are sufficient to identify NEXPs in modern artwork, e.g., intention and historical conditions. A recent model of the visual aesthetic experience

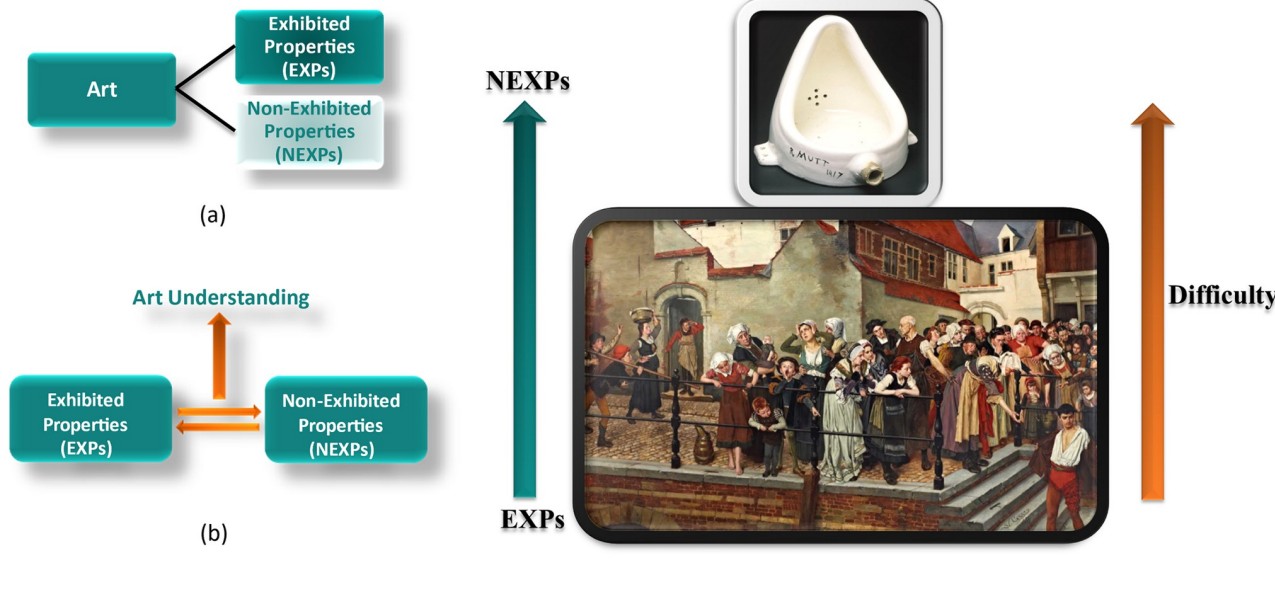

**Fig 1. Art analysis using Levinson's definition of art.** (a) Art consists of Exhibited Properties (EXPs) and None-Exhibited Properties (NEXP). (b) Art understanding is gained by relating EXPs to NEXPs rather than merely looking at EXPs. (c) The difficulty in art understanding is shown as a spectrum with an example for each end: top, "Fountain" by Marcel Duchamp, a conceptual piece with a greater significance of NEXPs, and bottom: "The Accident," by William Geet, a figurative piece with a more literal visual narrative and therefore more significance of EXPs.

suggests two parallel, quasi-independent processing modes: bottom-up perceptual processing which is universal among all humans (similar to EXP processing) and top-down cognitive processing that accounts for contextual information, artist intention, and artwork presentation circumstances (similar to NEXP processing) [10]. As summarized in Section Related Work, previous work suggests that DNN models can gain some insight on artwork meaning (semantics) starting only from EXPs, like color, texture, and shapes [2–4, 11, 12]. However, there are no comprehensive studies on the degree to which NEXP recognition can emerge during DNN training using artwork images, and whether such NEXPs are sufficient to distinguish art objects from non-art objects or other artwork, especially in case of conceptual arts. Such studies are important not only to identify and characterize the limitations of DNN models but also to understand if NEXPs of art objects can be sufficiently well distinguished using only their EXPs, thus if an art object is fully specified within its body of similar work, e.g., gallery or art show, or if NEXPs depend to a significant degree on elements not embodied into an art object, like contextual elements, the artist's intention, and the viewer's interpretation.

This paper presents a comprehensive experimental study on the degree to which a state-of-the-art Deep Convolutional Neural Network (DCNN) learns EXPs and NEXPs and then uses the learned knowledge to classify artwork into the same galleries and exhibitions as artists and curators did. As discussed in Section Related Work, existing work does not consider the boundary between the AI/ML's perception of a computer image and an image's interpretation as art. Thus, computational modeling of art is often not grounded in the theory of art. For example, AI/ML models are trained on art images labeled with their styles, genres, or authors but without information about their contexts, intentions, interpretations, or emotions. It is unknown the degree to which AI/ML models, like DCNNs, can automatically pick up these essential details. To address this problem, this work devised a new experimental study integrating semantic and conceptual ideas in aesthetics with AI/ML modeling and experimentation. Given that EXPs of art objects are the basis of DCNN model training while NEXPs are less likely to be learned well, two hypotheses were defined to ground the proposed study about the importance of EXPs and NEXPs in classifying artwork into different galleries:

*Hypothesis I*: The similarities and differences of EXPs within and between art galleries determine the difficulty level of classification using DCNN models.

*Hypothesis II*: DCNN models do not capture well enough the NEXPs needed for artwork classification into galleries. NEXPs do not influence the classification results.

A novel experimental method was devised to verify the two hypotheses. The method is grounded in a theoretical description of the art gallery classification problem. As EXPs also serve to describe NEXPs and hence their impact on classification is coupled, the method must separate the impact of EXPs from the impact of NEXPs, including the types of the relationships between EXPs and NEXPs and the contextual parameters. The method must also systematically describe the EXP and NEXP domains, so that experiments cover all defining situations. Using the similarity and dissimilarity of EXPs and NEXPs within and across galleries, the proposed approach identifies four situations in which the two impacts are distinguishable, and suggests three cases that express a gradual scaling of the coupling relationships between EXPs and NEXPs. Two novel ontologies were proposed to describe the EXP and NEXP domains in conceptual photography. The ontologies are grounded in the theory of art. Starting from gallery classification as a specific goal, the broader theoretical problem studied in this work is if computational methods can represent NEXPs only based on the EXPs of art objects, or whether subjective factors are essential, like the artist intention, viewer interpretation, and social context. This problem occurs in many situations that involve human input and subjective assessment, e.g., in humanities, social sciences, healthcare, education and so on, where implicit and tacit information plays a main role in deciding meaning.

Specifically, the devised method includes three experiments that used datasets assembled by an art expert. The used DCNN model was the VGG- 11 [13] pre-trained on ImageNet database [14]. The model was then retrained using art images. The datasets designed for the study included images of contemporary photography of diverse styles and conceptual orientations. Our art expert chose exhibitions of artists from different countries and photographs that reflect different approaches toward fine art photography, like realism, abstraction, commercial, and conceptual photography. As conceptual art, like Duchamp's "Fountain", often represents ordinary ready-made or mass-produced objects (or their photographs) as a work of art, and which exclusively relies on the ideas intended towards the artwork rather than the artistic style, the two hypotheses suggest that including conceptual photography increases the difficulty of classifying a dataset into galleries. A high EXP diversity within a gallery also increases the difficulty level. To further explore the degree to which the model learns NEXPs beyond EXPs, the study added a gallery of non-art images of ordinary objects that resembled in their appearance to the conceptual fine art photography exhibitions included in the experiment.

The two hypotheses indicate that the model should have a poor performance in this case because the high EXP similarity of the conceptual photography and non-art images. The experimental results were analyzed using statistical and classification metrics. The results of the three experiments confirmed the validity of the two hypotheses.

The paper has the following structure. Related work was summarized next, followed by the presentation of the theoretical description of the gallery classification problem and then the experimental methodology. Results and their discussion were described next. The paper ends with conclusions and further research directions.

## Related work

Recent work has proposed modern AI/ML methods to automatically analyze artwork for style recognition, classification, and generation [1–4]. A comprehensive overview paper discusses recent computational and experimental advances in visual aesthetics [15]. The AI/ML methods often use DCNNs, a DNN type devised for computer image processing [2–4, 11]. To address the need of large datasets for DCNN training [16], which is often difficult to meet for art, the traditional solution pre-trains a DCNN using large databases of images, e.g., ImageNet, and then retrains only the output and intermediate layers using art images [2–4, 11, 12, 17].

Automated style recognition attempts to identify the artistic style of art objects, like paintings and porcelain objects [2–4, 11, 12]. This work uses EXPs, like color and texture. For example, [18] examines the classification of artistic styles into their respective historical periods based on the ideas of Heinrich Wölfflin, a prominent art historian (1846–1945). Wölfflin explains that different artistic styles reveal their respective historical contexts. Therefore, a machine could classify artwork into historical periods by relying on the stylistic characteristics of artwork [18].

Also, different edge orientations are characteristic to traditional artwork from different cultures [19]. Statistical differences in image composition are presented between traditional art, bad art, and twentieth century abstract art [20]. Seven DCNN models were tested for three art datasets to classify genres (i.e. landscapes, portraits, abstract paintings, etc.), styles (e.g., Renaissance, Baroque, Impressionism, Cubism, Symbolism, etc.), and artists [2]. Classification uses mostly color information and achieves for some styles a recognition accuracy similar to human experts. However, certain styles are hard to be automatically differentiated from each other, like Post Impressionism and Impressionism, Abstract Expressionism and Art Informel, or Mannerism and Baroque [3]. A dual-path DCNN model recognizes both artistic style and

painting content [21]. DCNNs are proposed to recognize non-traditional art styles too, like Outsider Art style [12].

Work to uncover semantic information about art stems from the goal to understand the content of art objects, including the orientation of an object, the objects in a scene, and the central figures of a scene [13, 22–24]. Only EXPs are used in this work. Object orientation, e.g., if a painting is correctly displayed, uses low-level features, like simple, local cues, image statistics, and explicit rules [6, 22, 25, 26]. For example, using low-level features to train DNNs has been reported to be as effective as human interpretation across different granularities and styles [22]. The method performs better for portrait paintings than for abstract art, as portraits arguably include more reliable and repetitive cues, which improves DCNN learning.

Distinguishing image classes seems to focus on localized parts of a few, large objects. Low intra-class variability of the parts is important in those parts being learned. Different semantic parts might be selected for objects of related classes, like wheels for cars and windows for buses. Generative Adversarial Networks (GANs) are suggested for hierarchical scene understanding [27]. Analysis shows that the early layers learn physical positions, like the spatial layout and configuration, the intermediate layers detect categorical objects, and the latter layers focus on scene attributes and color schemes.

DCNN have been also used to recognize an artist that authored an artwork from a group of possible artists by learning the artist-specific, visual features (hence EXPs) of his/her work [4]. During DCNN training, various regions of an art image are occluded, so that the sensitivity of that region for correct classification can be established [4].

Experiments suggest that artist recognition uses low-level features, like material textures, color, edges, and the empty areas used to create visual patterns [4]. Other work advocates for using intermediate-level features, like localized regions, and some semantic features, e.g., scene content and composition [5]. Performance decreases if the pool of possible artists increases.

Fig 2 summarizes the existing work on various automated art understanding tasks.

Finally, artistic activities are inherently creative, creativity being a main research topic in psychology, neuroscience, sociology, organization research, and engineering design. Various methodologies have been proposed to aid creative work by focusing on topics, like concept

**Fig 2. Computational art understanding tasks.** NEXPs are more critical as more artistic aspects must be considered (i.e. fewer NEXPs are needed for style recognition and more NEXPs for gallery recognition).

formation, memory recall, concept combination, fixation, constraints, and analogies, to name just a few [28–32]. For example, constraints are critical in engineering design but they play a main role in artistic creativity too, like the way constraints shaped the work of Piet Mondrian [33] and Impressionist painters [34]. Existing software tools, including AI/ML methods, rarely consider insight from psychology or art history.

They effectively utilize EXPs, i.e., physical features (e.g., color, space, texture, form, and shape), principles of art (like movement, unity, harmony, variety, balance, contrast, proportion, and patterns), and subject topics (i.e. composition, pose, brushstrokes, and historical context). However, existing tools pay much less attention to capturing implicit and semantic information for situations in which subjective intention and perception play an important role in defining an object or a group of objects. Previous work suggests that DCNNs might learn some facets of the EXPs—NEXPs relationships, like the importance of visual cues, hierarchical compositions, and hidden structures [1, 5, 22, 23, 27], but it is unclear to what degree learning happens for situations in which NEXPs are the principal features in defining the meaning of artwork. This work attempts to address this gap in knowledge.

## Theoretical description

The problem of automated art gallery recognition can be described as the problem of identifying sets $\{\mathcal{M}(AO_i)|AO_i \in G_j\}$, where $\mathcal{M}(AO_i)$ is the intended message of the art object $AO_i$ in Gallery $G_j$, so that the message similarity of the artwork in any gallery is maximized and reduced across different galleries. The similar messages $\mathcal{M}$ of the art objects in a gallery correspond to the theme of the gallery.

The meaning of an art object $AO_i$ is captured by the following qualitative equation:

$$\mathcal{M}(AO_i) = \mathcal{M}(\{EXP_k^{(AO_i)}\}, \mathcal{N}(\{EXP_k^{(AO_i)}\}, Cont^{(G_j)})) \tag{1}$$

where set $\{EXP_k^{(AO_i)}\}$ is the discret set of the EXPs of the art object $AO_i$, parameter $Cont^{(G_j)}$ describes the information on contextualization and interpretation used to create the gallery $G_j$, and function $\mathcal{N}$ expresses the forming of NEXPs using EXPs, contextualization, and interpretation.

As multilayer feedforward NNs are universal approximators of continuous functions $f(\chi, \mathbb{R}^d)$, where parameter $\chi$ is a compact subset of $\mathbb{R}^d$ [35], the broader theoretical question that was studied in this work was whether DCNNs can learn equations (1) for the art objects of galleries curated by human experts, including the parameters $Cont^{(G_j)}$ of each gallery and the function $\mathcal{N}$ for expressing NEXPs. If DNNs can approximate well enough equations (1), then there is a computational way to express NEXPs, the context, and intention in artwork using only the EXPs of the artwork used in training.

Studying the validity of hypotheses I and II requires analyzing the impact of EXPs and NEXPs on the effectiveness of the DCNN learning of equations (1) for the art objects selected for an art domain. Note that EXPs and NEXPs are coupled in Eq (1) through function $\mathcal{N}$ and the contextual information $Cont$. Hence, there are two requirements for hypotheses verification: (a) a way to separate the effect of EXPs on classification from the effect of function $\mathcal{N}$ and parameter $Cont$, and (b) a representation to theoretically describe the EXP and NEXP domains for the considered artwork style, so that the verification tackles all its situations.

The following methodology was devised to experimentally verify hypotheses I and II.

## Experimental methodology

To separate the impact of EXPs on artwork classification into galleries from the impact of NEXPs, the similarity values of EXPs and NEXPs must be distinguishable with respect to their expected impact. Otherwise, EXPs might mask NEXPs due to their coupling. Four situations emerge corresponding to (i) high EXP similarity and NEXP dissimilarity and (ii) high EXP dissimilarity and NEXP similarity within and across art galleries. Poor classification performance for case (i) across galleries and case (ii) within galleries, and good classification performance for case (i) within a gallery and case (ii) across galleries validate the hypotheses. Any other result invalidates them.

Moreover, three cases describe the gradual impact of function $\mathcal{N}$ and the contextual parameter *Cont* on the NEXPs: (i) learning models that can distinguish between objects with (i.e. art objects) and without NEXPs (e.g., non-art objects), hence distinguishing between objects with $\mathcal{N} \approx 0$ and with $\mathcal{N} \neq 0$ in Eq (1), (ii) learning models $\mathcal{M}(\{EXP_k^{(AO_i)}\}, \mathcal{N}_{G_j}(\{EXP_k^{(AO_i)}\}, Cont^{(G_j)}))$ in which $\mathcal{N}_{G_j}$ is well defined and $Cont^{(G_j)} \approx constant$ for gallery $G_j$, i.e. there is a single mechanism through which NEXPs are created from EXPs, and (iii) the general case when $\mathcal{N}$ can be ill-defined and ambiguous and the context information (*Cont*) can be variable for a gallery. If the two hypotheses are correct then NEXPs do not impact classification for any of the three cases, otherwise DCNNs might learn some NEXPs under certain conditions.

Second, due to their discrete nature and diverse meanings, the EXP and NEXP domains are ontologies that enumerate the possible values, their ways of composition into structures, and their meanings. Table 2 indicates the ontology for EXPs. It expresses EXPs along four dimensions: (i) medium and color, (ii) shape, form and texture, (iii) composition, and (iv) subject matter. Table 3 presents the ontology for NEXPs, which are defined along three dimensions: (i) context, (ii) intention, and (iii) meaning. The two ontologies represent the EXPs and NEXPs in equations (1).

The experimental methodology in Fig 3(a) addresses the above requirements in its four parts: experiment design, dataset design, dataset verification, and model evaluation. Experiment design identified three experiments to reflect the above situations needed to verify the

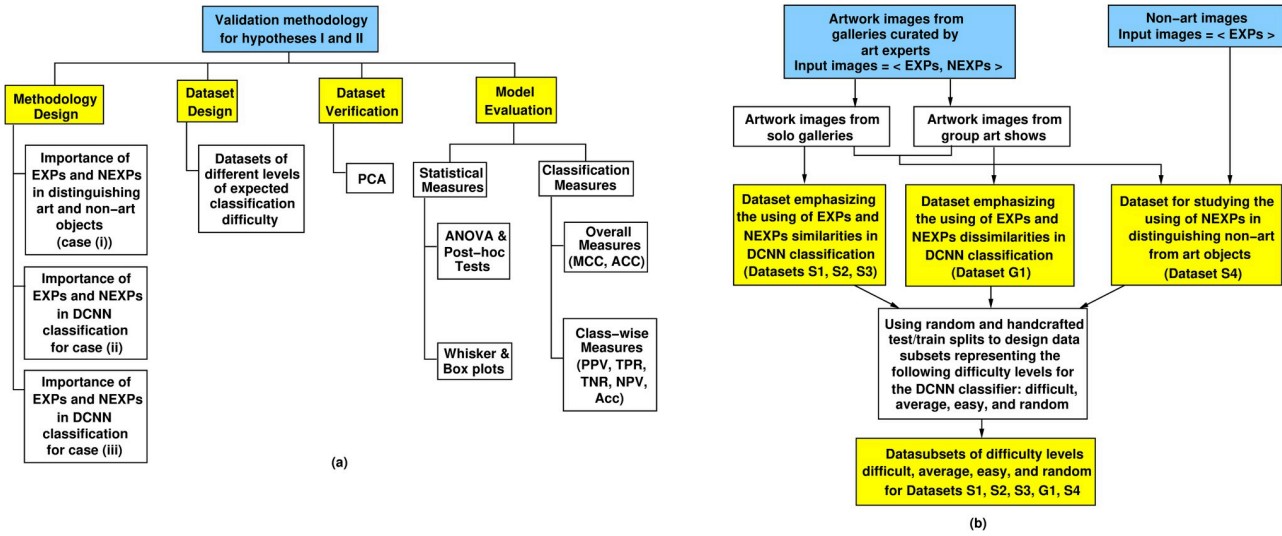

**Fig 3. Experimental methodology.** (a) The methodology to validate hypotheses I and II.(b) Dataset design method.

two hypotheses. Dataset design produced multiple datasets for DCNN model training and validation, as shown in Fig 3(b).

All datasets were verified with respect to their intended purpose for the study. A trained and fine-tuned DCNN model was used to classify the images of the datasets into galleries. The effectiveness of the DCNN model was evaluated using statistical measures and classification metrics. Artwork classification using the DCNN model and classification by human experts were also compared. The parts of the methodology are discussed next.

## Dataset design

The validity of the two hypotheses was verified by studying the DCNN gallery classification performance for artwork datasets with different mixtures of EXPs and NEXPs. Five datasets (called S1, S2, S3, G1, and S4) were put together to reflect different EXP and NEXP similarity and dissimilarity cases, as shown in Fig 4. The datasets capture the requirements presented in the previous section.

Specifically, the two hypotheses suggest that a dataset is harder to classify by the DCNN model if its images in galleries with different themes (hence, dissimilar NEXPs) are more similar with respect to their visual appearance, i.e. EXPs. In other words, higher similarity of EXPs across galleries increases the level of difficulty. For instance, Fig 5 shows two photos from two galleries in Dataset S1. One photo is from Gallery "Mukono" and the other is from Gallery "Heat + High Fashion". Although these photos were taken by two different artists from

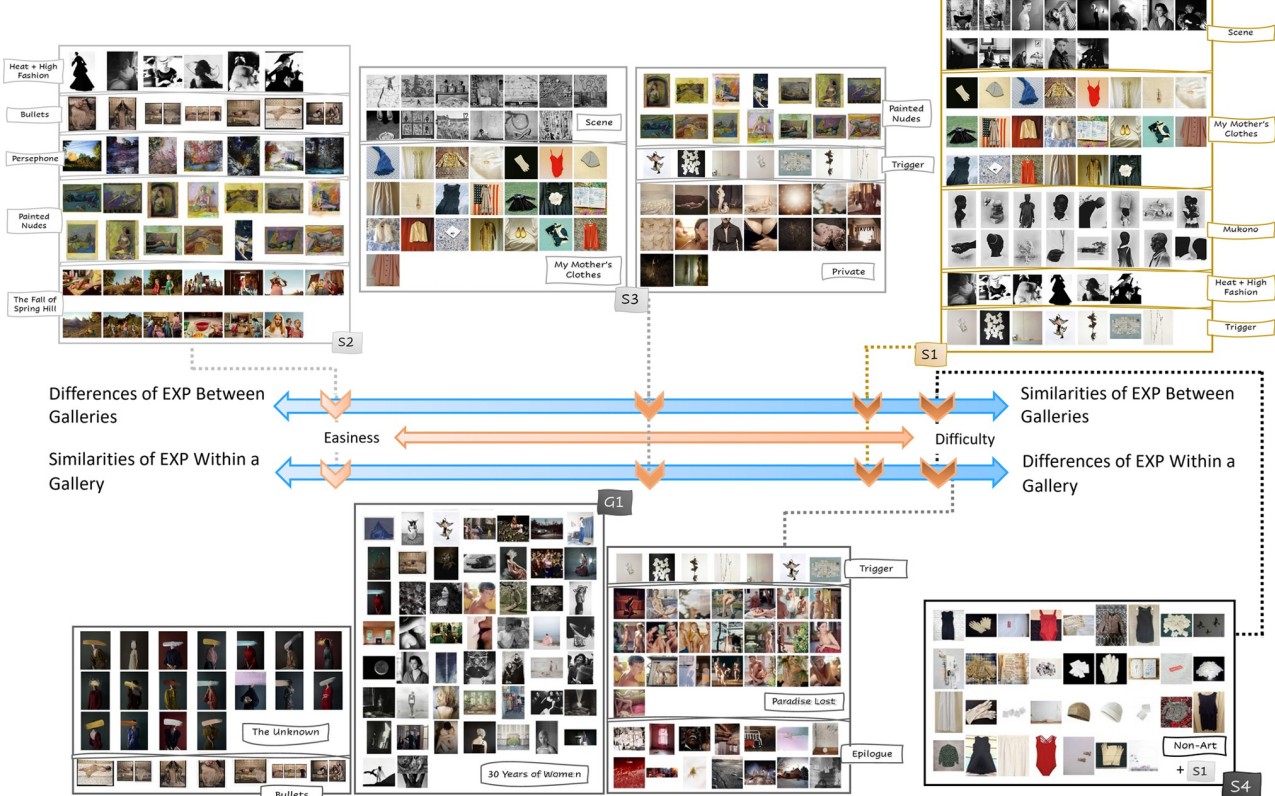

**Fig 4. Dataset summary.** The designed datasets S1, S2, S3, G1, and S4 pertain to a broad difficulty spectrum due to their EXPs and NEXPs dissimilarities and similarities within or across galleries.

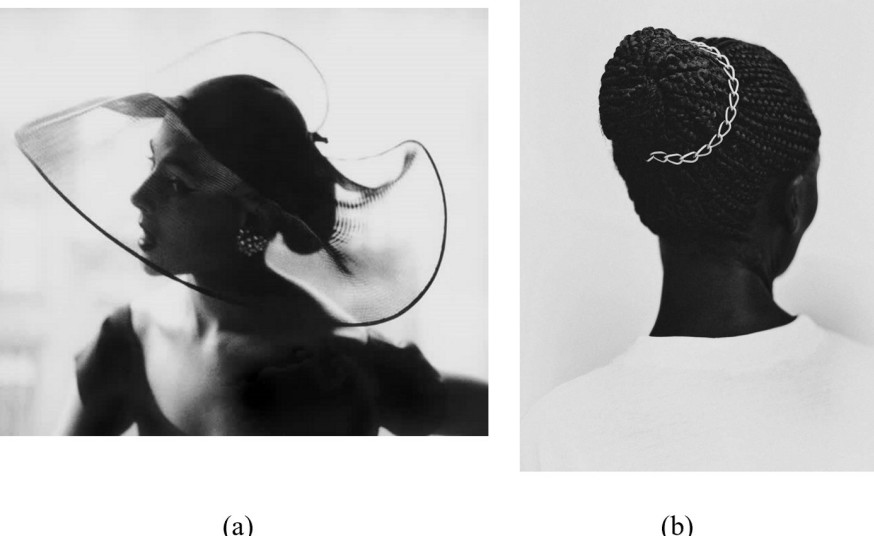

(a)                                                    (b)

**Fig 5. Individual examples from dataset S1.** (a) "Translucent Hat" from Gallery "Heat+ High Fashion". (b) "Hairpiece" from Gallery "Mukono". The two photos illustrate EXP similarities between the galleries of the dataset. Black and white photography and the high contrast of a dark figure on a light background are some EXPs that the two photos and their respective galleries have in common. These similarities between the galleries of a dataset increase the classification difficulty.

different contexts engaging different themes (one ethnicity and race, and the other female fashion in the 1950s), they are similar in their EXPs. Both are black and white photos with high contrast between the dark figure on a light background. Such similarities of EXPs between two or more galleries in a dataset would increase the difficulty. Dataset S1 was designed to capture this observation. Moreover, a high EXP diversity within a gallery makes image classification harder, even if the images of a gallery have similar NEXPs. Dataset S2 was created for this case. A third dataset, S3, has within and inter-gallery EXP and NEXP similarity values between the values for the datasets S1 and S2. Datasets S1, S2, and S3 were created using artwork images from solo galleries, as the NEXPs diversity of the art objects in a gallery is likely to be lower, as they share the same author (hence, $\mathcal{N}_{G_j}$ is well defined and $Cont^{(G_j)} \approx constant$ for gallery $G_j$, as previously explained).

Moreover, a high EXP dissimilarity within a gallery is expected to increase the classification difficulty even if the NEXP dissimilarity across galleries is high, i.e. due to different themes and authors, because it is harder for the DCNN model to find common EXPs for a gallery. As explained before, for group art exhibitions, function $\mathcal{N}$ can be ill-defined and ambiguous and the context information $Cont$ can be variable for the same gallery, as the galleries were curated around similar themes but contain art objects that look different from each other, as they are by distinct artists. Dataset G1 was assembled using images from group art shows.

Finally, a slightly different but related problem of using NEXPs in images classification is the separation of non-art images from art images, thus, separating images with $\mathcal{N} \approx 0$ from the images with $\mathcal{N} \neq 0$. Dataset S4 was assembled to test the capacity of the DCNN model to learn the presence or absence of NEXPs in images.

The limits of DCNN classification as compared to the classification by art experts were also studied. The assembled datasets were input to the DCNN model with random and handpicked test/train splits to create difficult, average, and easy classification situations. These splits

**Table 1. Characteristics of dataset S1: The related galleries (e.g., names of the exhibitions or shows), the number of images per gallery, and the total number of images per dataset.**

| Dataset | Related galleries | Number of Images | Total |
|---------|-------------------|------------------|-------|
| S1 | *Heat + High Fashion* | 6 | 64 |
| | *Mukono* | 16 | |
| | *My Mother's Clothes* | 22 | |
| | *Scene* | 13 | |
| | *Trigger* | 7 | |

targeted different mixtures of EXPs and NEXPs sets. Four versions, called *difficult*, *average*, *easy* and *random*, were created for each dataset.

Finally, the degree to which the dataset size influences gallery classification performance was studied using galleries with different numbers of images in a gallery.

The considered art galleries were chosen from existing online exhibitions curated by established art curators, and not by the art expert that participated to our study.

To limit the scope of the datasets, we picked artwork that uses photography as an underlying medium, so that the galleries reflect diverse approaches towards fine arts photography in contemporary art.

The artwork pursues different artistic attitudes, like highly conceptual, representational, and abstract. Certain art objects mix photography with paint, some used digital, and others are analog photography.

We also included group exhibitions, as group exhibitions were curated around a common theme or concept but are vastly different in their styles and formal features.

Table 1 summarizes the characteristics of the galleries in Dataset S1 (see the Appendix for the other datasets). Table 2 details the EXPs and Table 3 the NEXPs of the galleries in this dataset, where *composition* is the arrangement of visual elements within a frame, *subject matter* is what we are looking at, *context* is historical, social, political, cultural conditions in which the work is created, and *intention* refers to artists' intention to make a work of art with a meaningful connection to previous works of art and the history of art. Fig 6 shows the outliers of the galleries, such as the images that are different from the rest.

## Design verification

Principal Component Analysis (PCA) was used to verify the designed datasets with respect to their purpose for the experimental study. Each image in a dataset underwent the same pre-processing as the images used for pre-training the DCNN model (except for data augmentation transformations), such as resizing and center-cropping to a 224 × 224-pixel image in RGB (color images) and L (grayscale images) color space.

The first three principal components of each image were then plotted in a 3D scatter plot. Each gallery was shown using a separate color. The formation of distinct clusters with points of the same color indicates a successful image classification using the chosen features. Random placement of points of the same color indicates poor classification. The paper presents the PCA 3D plots for the Dataset S1 (Fig 7). Similar figures were shown for the other datasets in the supporting information part.

## DCNN model

The three experiments used a VGG-11 DCNN model [13] pre-trained on ImageNet dataset [14] and discriminatively fine-tuned [38]. The model came from PyTorch framework. To

**Table 2. EXPs of the galleries in dataset S1.**

| Gallery | EXPs | | | |
|---|---|---|---|---|
| | **Medium, Color** | **Shape, Form, Texture** | **Composition** | **Subject Matter** |
| *Heat + High Fashion* | • Black and white photography<br>• Monochromatic<br>• Dark value<br>• Light value<br>• Midtones<br>• High contrast | • Figurative<br>• Organic<br>• Plain<br>• Open forms<br>• Painterly | • Closed compositions<br>• Open composition<br>• Tendency toward symmetry<br>• A-symmetrical<br>• Centered<br>• Alignment of the subject matter<br>• Horizontal frames<br>• Vertical frames<br>• Emphasis on a single subject matter<br>• Empty and quiet composition<br>• Busy and crowded compositions | • Female body<br>• Female torso<br>• Portraits<br>• Hidden human faces<br>• Fashion<br>• Interior space<br>• Shadows, reflections |
| *Mukono* | • Black and white photography<br>• Monochromatic<br>• Dark value<br>• Light value<br>• Midtones<br>• High contrast<br>• Medium contrast<br>• Low contrast<br>• Low saturation<br>• Neutral colors | • Figurative<br>• Organic<br>• Plain<br>• Open forms<br>• Closed forms | • Closed composition<br>• Tendency toward symmetry<br>• Asymmetrical<br>• Centered alignment of the subject matter<br>• Horizontal frame<br>• Vertical frame<br>• Emphasis on a single subject matter<br>• Empty and quiet composition | • Human torso/male torso<br>• Female torso<br>• Portraits<br>• Hidden human faces<br>• Nature/landscape<br>• Human body<br>• Everyday objects<br>• Still life<br>• Animals |
| *My Mother's Clothes* | • Color photography<br>• High saturation<br>• Medium saturation<br>• Low saturation<br>• Cool colors<br>• Warm colors<br>• Neutral colors<br>• Dark value<br>• Light value<br>• Midtones<br>• High contrast<br>• Low contrast<br>• Medium contrast<br>• Chromatic | • Organic<br>• Geometric<br>• Textured<br>• Plain<br>• Open form<br>• Linear<br>• Closed form<br>• Decorative<br>• Pattern<br>• Floral<br>• Text | • Closed composition<br>• Tendency toward symmetry<br>• Asymmetrical<br>• Centered alignment of the subject matter<br>• Square frames<br>• Emphasis on a single subject matter<br>• Busy and crowded compositions<br>• Empty and quiet composition | • Female clothes<br>• Still life<br>• Everyday objects<br>• Domestic space |
| *Scene* | • Black and white photography<br>• Monochromatic<br>• Dark value<br>• Light value<br>• Midtones<br>• High contrast<br>• Medium contrast | • Figurative<br>• Textured<br>• Plain<br>• Open form<br>• Closed-form<br>• Pattern | • Closed composition<br>• Tendency toward symmetry<br>• Asymmetrical<br>• Centered alignment of the subject matter<br>• Square frame<br>• Emphasis on a single subject matter | • Human body<br>• Female body<br>• Male body<br>• Human torso<br>• Male torso<br>• Female torso<br>• Interior space<br>• Shadows, reflections<br>• Artists |

*(Continued)*

**Table 2.** (*Continued*)

| Gallery | EXPs | | | |
|---|---|---|---|---|
| | **Medium, Color** | **Shape, Form, Texture** | **Composition** | **Subject Matter** |
| *Trigger* | • Color photography<br>• Medium saturation<br>• Low saturation<br>• Neutral colors<br>• Cool colors<br>• Dark value<br>• Light value<br>• Mid-tones<br>• High contrast<br>• Low contrast<br>• Medium contrast<br>• Chromatic | • Plain<br>• Organic<br>• Geometric<br>• Closed form | • Closed composition<br>• Tendency toward symmetry<br>• Asymmetrical<br>• Centered alignment of the subject matter<br>• Horizontal frame<br>• Vertical frame<br>• Emphasis on a single subject matter<br>• Empty and quiet composition | • Interior space<br>• Domestic space<br>• Still life<br>• Animals |

**Table 3. NEXPs of the galleries in dataset S1.**

| Gallery | NEXPs | | |
|---|---|---|---|
| | **Context** | **Intention** | **Meaning** |
| *Heat + High Fashion* | • Modern and contemporary<br>• New York"bohemian." | • Engaging female fashion through photography<br>• Realism yet a sense of ambiguity through blurred images and painterly qualities of the medium of photography (in that sense, it can place itself against modernist photography and its notion of medium specificity) | • Photographs of dancer Isadora Duncan<br>• Fashion photography that reflects the time and historical context<br>• Female identity through fashion and clothing<br>• Realism yet a sense of ambiguity through blurred images and painterly qualities of the medium of photography |
| *Mukono* | • Contemporary photography<br>• Cultural studies | • Realism and documentary photography | • Documenting people around the world<br>• Race, ethnicity, culture-racial and cultural identity |
| *My Mother's Clothes* | • Contemporary photography conceptual art (using ready-made objects as works of art) | • Conceptual photography inspired by conceptual art and the use of readymade/ordinary objects and blurring the boundary between art and life<br>• Photographing her mother's clothes and personal items as a form of portraits or chronology of her mother's life [36]<br>• Using art and photography to cope with the loss of her mother and her mother's suffering from Alzheimer [36] | • Her mother's clothes and personal items as her mother's portrait/body = clothes as a metonymy of the person<br>• Remembering the past, memories of her mother, perhaps a sense of nostalgia<br>• Coping with the trauma of losing her mother and her suffering from Alzheimer<br>• Gender expression/identity<br>• Social class in America |
| *Scene* | • 1960s underground /avant-garde artists' scenes in the United States | • Realistic photographs of Avant-garde artists in New York during the 1960s<br>• Documentary photography | • Photography and realism<br>• Indexicality<br>• Representation of avant-garde artists in NYC during the 1960s<br>• Human emotion and psychological expression |
| *Trigger* | • Contemporary photography<br>• Conceptual photography | • Conceptual photography<br>• Photographing everyday objects and domestic space (her hometown)<br>• Capturing time passing through photography | • Artist's hometown and the lives of people who lived there [37].<br>• Passage of time [37]<br>• Her personal experiences [37]<br>• Collision of past and present [37]<br>• Domestic space and meaningful—perhaps personal everyday objects [37]<br>• Time, temporality |

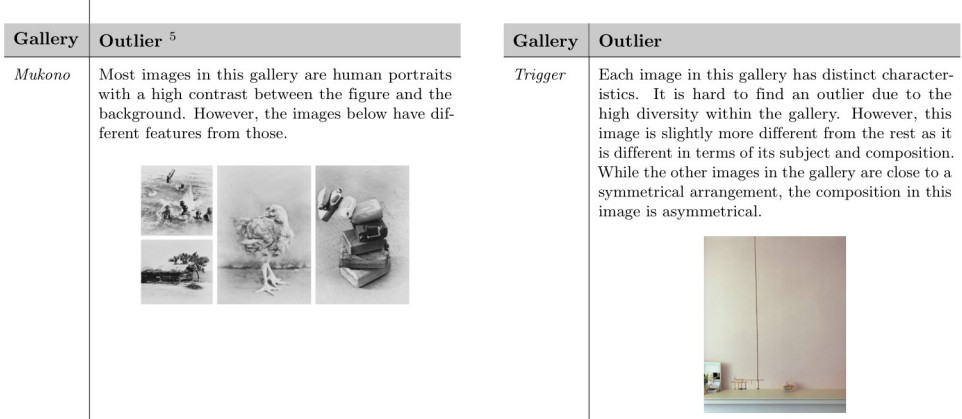

| Gallery | Outlier [5] |
|---|---|
| *Mukono* | Most images in this gallery are human portraits with a high contrast between the figure and the background. However, the images below have different features from those. |

| Gallery | Outlier |
|---|---|
| *Trigger* | Each image in this gallery has distinct characteristics. It is hard to find an outlier due to the high diversity within the gallery. However, this image is slightly more different from the rest as it is different in terms of its subject and composition. While the other images in the gallery are close to a symmetrical arrangement, the composition in this image is asymmetrical. |

**Fig 6. Gallery outliers for dataset S1.** Galleries *Heat + High Fashion, My Mother's Clothes*, and *Scene* do not have any outliers.

avoid overfitting, batch normalization [39] was used as a regularization technique along with data augmentation using methods *Random Rotation*, *Random Horizontal Flip*, and *Random Crop with Padding* from Torchvision library [40]. The learning rate obtained by method *Cyclical Learning Rates* [41] for the transferred features was an order of magnitude lower than that

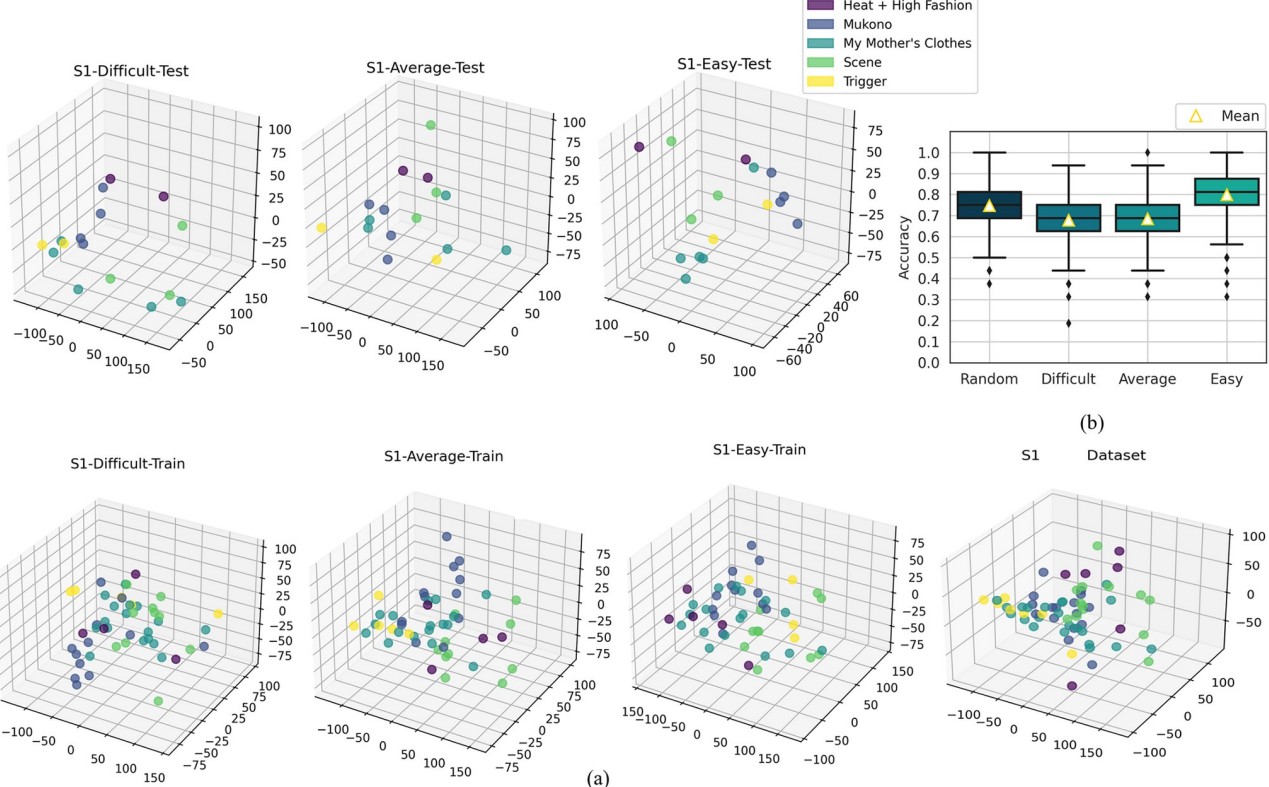

**Fig 7. Results for dataset S1.** (a) PCA plots for the training and test data of subsets *difficult*, *average*, and *easy*. (b) Box and whisker plots of the overall accuracies (ACC) of the DCNN classification of the four subsets. Subsets *difficult* and *average* have outliers of 20% and 100%.

of the output classifier layer. The DCNN model training utilized Adam Optimizer [42] with a cross-entropy loss function.

## Model evaluation

The trained DCNN model was tested using multiple statistical measures and classification metrics [43]. Statistical measures assessed the model behavior in response to the intended experimental setup. Unimodal distributions indicate a single behavior of the model, and multimodal distributions show if the model responds in multiple ways or if undetected, underlying variables are present. The comparison of means through ANOVA tests can identify interpretable characteristics of the model's responses to the designed inputs. Classification metrics evaluate a model's response to a specific task, and facilitate an unbiased and more balanced assessment of the DCNN model.

**Statistical measures**. A total of 1400 trials were collected for each experiment to sample the space of possibilities. Unique seeds to torch and all other random processes were used to maximize the likelihood of finding outliers.

A one-way ANOVA test with a confidence interval of 0.99 ($\alpha$ = 0.01) was performed as the primary measure for statistical comparison of the model's overall performance. Although ANOVA test is robust to the non-normality of the distribution and to some degrees of the heterogeneity of variances with equal sample sizes [44, 45], we also performed Leven's test of homogeneity of variances [46] and Shapiro-Wilk normality test [46], and visually verified these assumptions by assessing the histograms and normal Q-Q plots. For large sample sizes, like ours, minuscule derivations from normality can be flagged as statistically significant by parametric tests [47–49], which suggests the need to visually inspect the distributions. To pinpoint the different pairs and to consider the deviations from normal distributions of homogeneous variance when using ANOVA test for comparison of means, Games-Howell (nonparametric) [50] and Dunnett's T3 (parametric) [51] post-hoc tests were carried out to account for the violation of homoscedasticity or equality of variances. Tukey's test [52] was performed to control Type I errors or the likelihood of an incorrect rejection of the hypothesis. All statistical tests were conducted using SPSS. Plots were created using seaborn API.

**Classification metrics**. Eight class-wise measures were computed to observe the DCNN model's performance for each art gallery: Positive Predictive Value (PPV, Precision), True Predictive Rate (TPR, Recall, Sensitivity), True Negative Rate (TNR, Specificity), Negative Predictive Value (NPV), False Positive Rate (FPR), False Negative Rate (FNR), False Discovery Rate (FDR), and Class-wise Accuracy (Acc). In addition to the overall accuracy (ACC), Matthew's Correlation Coefficient (MCC) was utilized to avoid overemphasized (inflated) results [53]. For brevity of the presentation, only the measures primarily evaluating True values (True Positive, True Negative), PPV, TPR, TNR, and NPV were used to analyze the DCNN model's performance. The full report of the discussed statistical measures and plotted results were presented in the supporting information section.

## Experiments

### Experiment I. The impact of EXPs and NEXPs in classifying solo art shows

**Dataset S1 description.** As explained in Subsection Dataset Design and summarized in Tables 2 and 3, Dataset S1 was designed by our art expert to have the the most similar EXPs and most dissimilar NEXPs among its galleries as compared to the datasets S2 and S3. If hypotheses I and II were true, then these EXP and NEXP characteristics among galleries result in a poor classification performance of the DCNN model. Good classification performance rejects at least one of the two hypotheses.

**Results**. The PCA 3D plots in Fig 7(a) depict the similarity of the art images of the three subsets *difficult*, *average*, and *easy* built for Dataset S1 using different test/train spits, as explained in Subsection Dataset Design. Different colors indicate different galleries. Comparing the plots for subsets *easy*, *average*, and *difficult*, the gradual formation of single-color clusters of points can be noticed, with fewer occlusions and mixtures of images from different galleries. However, there were no fully homogeneous clusters. The PCA plots have the three principal components variance of 79.5%, 67.82%, and 54.44% for the three subsets. This result validates their purpose intended for studying the impact of the training/test sets on the DCNN performance.

The performance results obtained for classifying Dataset S1 into galleries using the DCNN model were as follows. ANOVA post-hoc tests indicated that there was no statistically significant difference between the ACC of the subsets *difficult* and *average* ($p_{Tukey\ HSD}$ = 0.236, 99% C.I = [-0.0049, 0.0199], $p_{Dunnett\ T3}$ = 0.374, 99% C.I = [-0.0057, 0.0207], $p_{Games-Howell}$ = 0.283, 99% C.I = [-0.0056, 0.0206]), even though the box plots in Fig 7(b) and the bar plots of the class-wise metrics in Fig 8 suggested otherwise. ANOVA post-hoc tests for the rest of the subset pairs showed a statistically significant difference ($p < 0.01$ for all tests). Hence, the DCNN model had a distinct behaviour for each of the subsets.

Next, the capacity of the DCNN model to correctly identify a specific gallery was studied for the subsets *random*, *easy*, *average*, and *difficult* of Dataset S1. The class-wise metrics in Fig 8 displayed distinct distributions for the four subsets. The closeness of MCC averages (i.e. 0.69, 0.61, 0.62, and 0.76) and ACC averages (e.g., 0.75, 0.68, 0.68, and 0.80) for the subsets *random*, *difficult*, *average*, and *easy* suggest a low possibility of random assignment of gallery labels by the model. However, metric Acc was not reliable in understanding the model's ability to find the correct gallery labels, as it barely changed for any gallery. More insight was obtained by analyzing the other metrics. As NPV and TNR measure True Negatives (TNs), their relatively continuous high values suggest that the DCNN model was relatively successful in indicating that a certain artwork image is or is not part of a given gallery. PPV and TPR measure True Positives (TP), hence the DCNN's ability to identify the correct gallery of an artwork

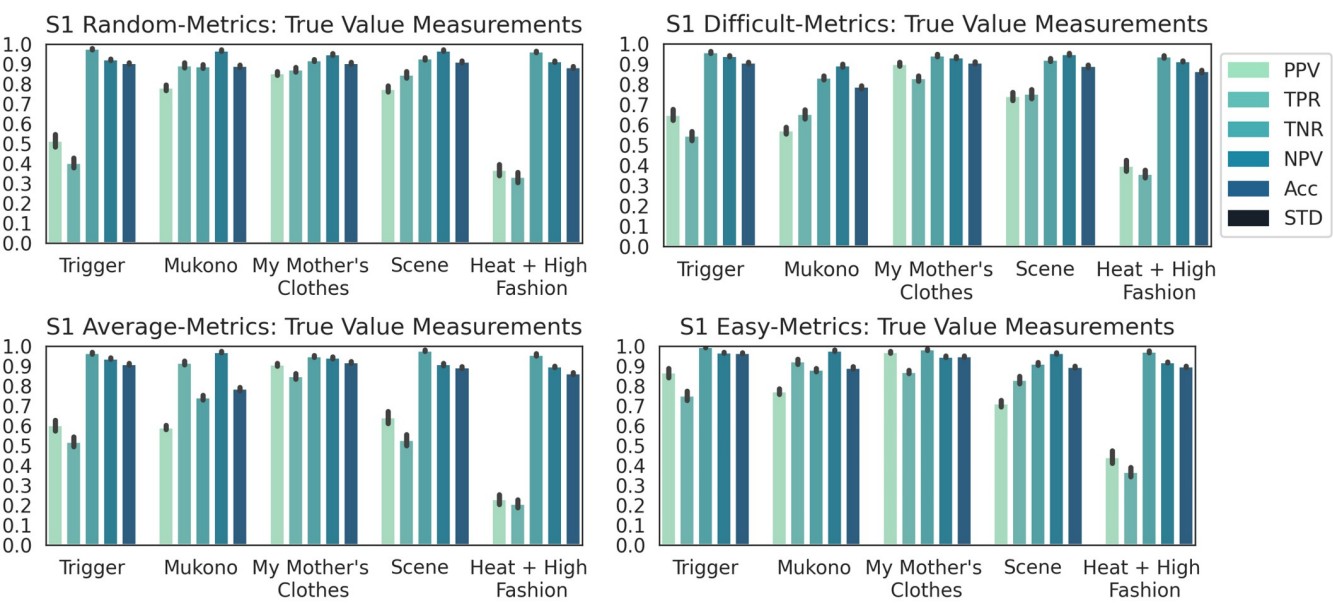

**Fig 8. Class-wise classification metrics results for dataset S1.**

image. As Fig 8 shows, PPV and TPR were consistently low for galleries "Trigger" and "Heat + High Fashion", and low for subsets *difficult* and *average* of galleries "Scene" and "Mukono". PPV and TPR were consistently high only for Gallery "My Mother's Clothes" and for subsets *random* of galleries "Mukono" and "Scene". Hence, with the exception of Gallery "My Mother's Clothes", the DCNN model struggled with finding the correct gallery of the artwork images. An additional experiment was performed to clarify whether the high performance obtained for Gallery "My Mother's Clothes" was due to EXPs or NEXPs being learned by the model in this case. The experiment, called Experiment III, was discussed in a separate subsection. In this experiment, additional images were added as part of a new gallery called "Non-Art", so that these images were very similar in their EXPs with the galleries "Trigger" and "My Mother's Clothes" but had no NEXPs, as they are not artwork. As shown in Fig 12, the new gallery worsened the classification performance, which suggests that the DCNN model did not learn NEXPs and was negatively affected by the increased EXP similarity between distinct galleries. Thus, the experiments using Dataset S1 confirmed the two hypotheses.

A detailed analysis of the results was then performed by the art expert to understand how EXPs and NEXPs influenced the DCNN classification as compared to human classification into galleries. Even though Gallery "Trigger" was expected to be the hardest to be automatically classified among all galleries, it was actually the second hardest. Instead, Gallery "Heat+ High Fashion" produced the lowest performance, likely because it is one of the three grayscale galleries. It shares similar EXPs with Gallery "Scene", and its size is slightly smaller than the other two galleries (see the supporting materials section). Assuming that the model learned NEXPs and used them for classification, distinguishing the three grayscale galleries with dissimilar NEXPs (due to their historical differences) would be easier. However, this situation was not observed.

As summarized in Table 2, Gallery "My Mother's Clothes" had the most diverse EXPs compared to the other galleries (and the most similar EXPs within the gallery), which explains the high performance in correctly finding the gallery for its images. For the situations with a high TP performance, the values were similar for subsets *random* and *easy*. Thus, randomly selecting training images for these situations is likely to include enough diverse EXPs to support a relatively correct classification, as estimated by our art expert. Moreover, DCNN performance was less linked to NEXP diversity.

Cross-referencing PCA and the class-wise metrics showed that the inter-class similarities of principal components represented by Euclidian distance in a 3D space pose more challenges than the within-class dissimilarities. This was observed in three instances for Dataset S1: (i) The performance for Gallery "Heat+ High Fashion" was the lowest, even though two test images in subset *average* were close to each another. This is likely because other classes' datapoints were concentrated nearby. (ii) The low performance for Gallery "Trigger" is due to the occluded points in subset *difficult* and the distant points in subset *average*.

The best performance was obtained for subset *easy*, as there were no points from other classes present between the datapoints of the two test images. (iii) Gallery "Scene" 's performance was the lowest for subset *average* and about similarly high for the other subsets, likely due to the placement in close proximity of its three test images as well as the closeness of subset *average*'s points to the other galleries. The gallery size was not critical on its own in setting the difficulty level of a gallery, however, it biased the classifier in some cases towards the larger galleries.

**Dataset S2 description**. Dataset S2 was designed by our art expert to have the most dissimilar EXPs and most similar NEXPs between galleries compared to the datasets S1 and S3. If hypotheses I and II were true, then these EXP and NEXP characteristics should produce a strong DCNN classification performance. A poor classification performance rejects at least one of the two hypotheses.

**Results**. The PCA 3D plots for Dataset S2 were included as supporting information, and they confirmed the intended purpose of the dataset for the experiments.

The performance for classifying Dataset S2 into galleries using the DCNN model was as follows. ANOVA post-hoc tests showed a statistically significant difference between the ACC values of all subsets ($p < 0.01$ for all tests). Although the box plots (in the supporting information) of subsets *random* and *average* were almost identical, the two subsets were still distinguishable by their outliers and the class-wise metrics in Fig 9. Hence, the DCNN model had a distinct behavior for each subset.

The capacity of the DCNN model to correctly identify a specific gallery was studied using the four subsets of Dataset S2. The class-wise performance metrics in Fig 9 displayed distinct distributions for the subsets. The small changes of metric Acc suggest that the metric is unreliable in finding the correct gallery labels. The closeness of MCC averages (i.e. 0.92, 0.80, 0.94, and 0.99) and ACC averages (e.g., 0.93, 0.82, 0.94 and 0.99) for subsets *random*, *difficult*, *average*, and *easy* indicate a low possibility of random assignment of gallery labels by the model. The classification performance was strong for all galleries, which supports the validity of hypotheses I and II.

The detailed analysis of the other metrics produced the following observations. Subset *easy* had almost perfect values for all metrics and for all galleries. Slightly worse, Subset *average* had close to perfect metric values for all galleries with the exception of galleries "Painted Nudes" and "Heat + High Fashion". The relatively low value of TPR for Gallery "Painted Nudes" indicates that False Negatives (FNs) are the cause of the worse performance, as the DCNN model did not recognize well the images in this gallery.

False Positives (FPs) reduced the classification performance for galleries "Heat + High Fashion" and "Fall of Spring Hill", i.e. the DCNN model mistook images in Gallery "Painted Nudes" as being in the two galleries. For subset *random*, the DCNN model struggled with galleries "Persephone" and "Bullets". It misclassified their images as being in Gallery "Painted Nudes". Galleries "Heat + High Fashion" and "The Fall of Spring Hill" had the lowest and second lowest PPV and TRP values. For subset *difficult*, the images in galleries "Fall of Spring Hill" and "Bullets" were hard to classify.

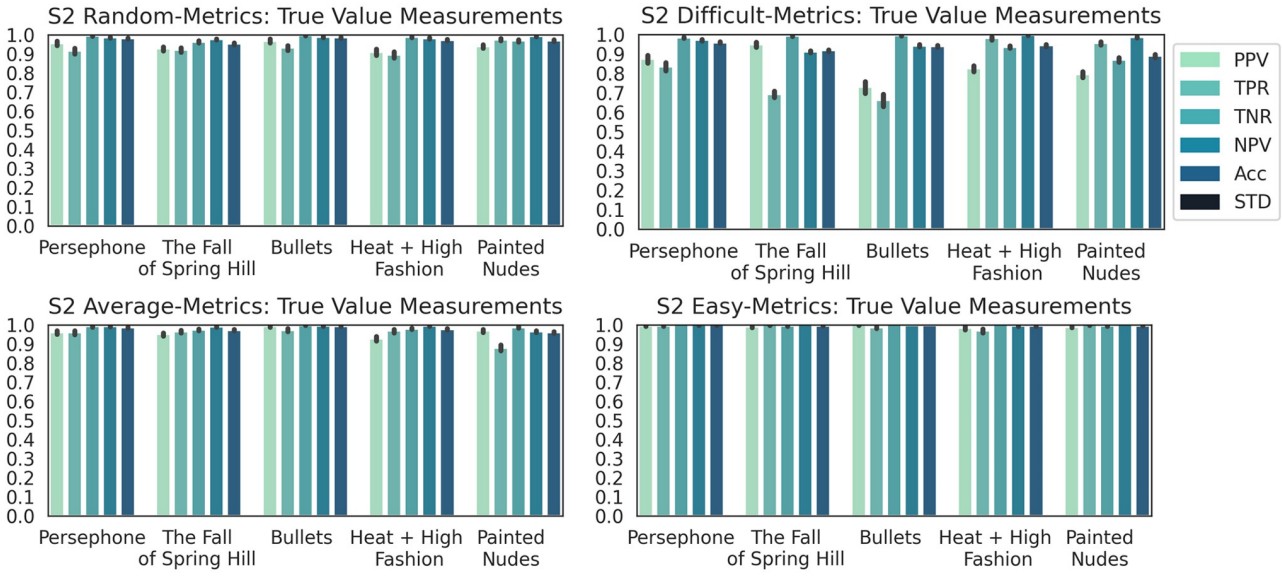

**Fig 9. Class-wise classification metrics results for dataset S2.**

The model confused images in the two galleries as being in galleries "Heat + High Fashion" or "Painted Nudes" likely because of the many human figures in these galleries.

**Dataset S3 description.** Dataset S3 was designed by our art expert to have the EXP and NEXP similarities and dissimilarities within and across galleries between those of the datasets S1 and S2.

If hypotheses I and II were true then having similarities and dissimilarities in-between those for datasets S1 and S2 would result in a classification performance for Dataset S3 that is between those for the two datasets. A strong or a poor classification performance for Dataset S3 rejects at least one of the hypotheses.

**Results**. The PCA 3D plots for Dataset S3 were provided as supporting information. The plots confirmed the desired purpose of the dataset for the experiments.

The obtained performance for classifying Dataset S3 into galleries was presented next. ANOVA post-hoc tests showed a statistically significant difference between the ACC of all subsets ($p < 0.01$ for all tests). The box plots (in the supporting information) of all subsets indicated three distinct distributions. Hence, the DCNN model had a distinct behavior for each of the subsets.

The capacity of the DCNN model to correctly identify a specific gallery was studied using the four subsets of Dataset S3. The class-wise metrics in Fig 10 displayed distinct distributions for the four subsets. The small changes of metric Acc suggest that it is unreliable in finding the correct gallery labels. The closeness of MCC averages (i.e. 0.73, 0.53, 0.68 and 0.84) and of ACC averages (e.g., 0.77, 0.62, 0.74 and 0.869) for subsets *random*, *difficult*, *average*, and *easy* indicate a low possibility of random assignment of gallery labels by the model. TNR and NPV values are high for all situations, hence the DCNN model can often correctly indicate if an artwork does not pertain to a gallery. PPV and TPR values are superior than for Dataset S1 but lower than for Dataset S2. These results support the two hypotheses.

The detailed analysis of the metrics showed that the DCNN model consistently succeeded in classifying galleries "Boarding House" and "Painted Nudes" for all their subsets. This was due to the high EXP dissimilarity of the two galleries and the rest of the dataset. Gallery

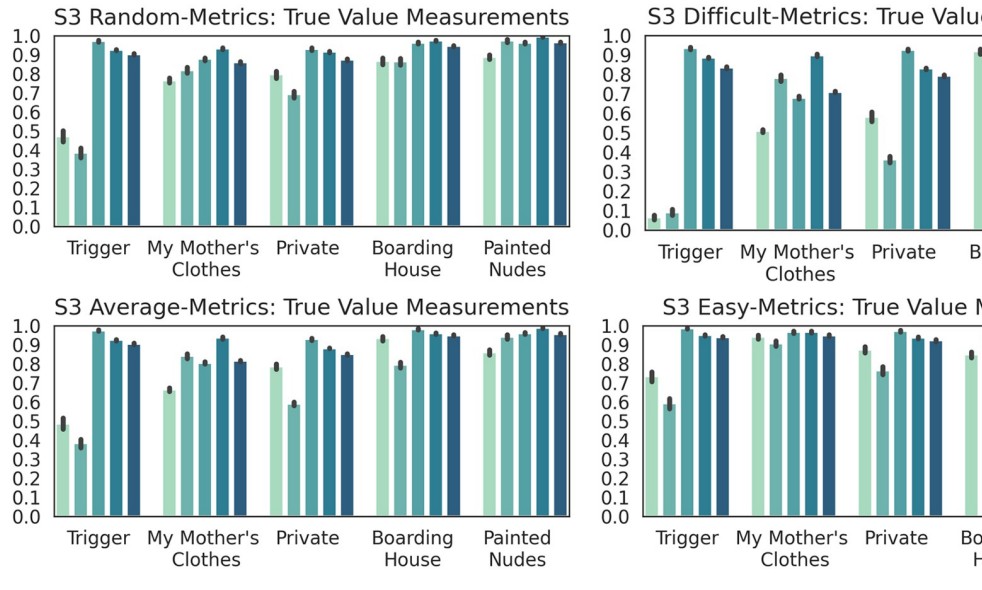

**Fig 10. Class-wise classification metrics results for dataset S3.**

"Boarding House" was the only grayscale gallery, and "Painted Nudes" was the only mixed medium gallery with textures of paint and brush on top of photography.

The low performance for all subsets of Gallery "Trigger" was because of its many NEXPs, as the gallery presents conceptual art. For the cases with a low performance, like Gallery "Trigger" and subset *difficult* of Gallery "Private", TPR values were often less than PPV values, hence, the artwork in these galleries were incorrectly assigned to other galleries. For example, images in galleries "Private" and "Boarding House" were assigned to Gallery "My Mother's clothes".

### Experiment II. The impact of EXPs and NEXPs in classifying group shows

**Dataset G1 description**. This experiment investigated the impact of the within-gallery EXP diversity on the DCNN classification performance while the NEXPs similarity was high for each gallery. To that end, Dataset G1 was designed by our art expert to include two group exhibitions by multiple artists with distinctive styles, hence diverse EXPs, while their artwork shared NEXPs that allowed the curator to assemble them in a single group exhibition. If hypotheses I and II were true then the classification performance for Dataset G1 should be worse than the performance in Experiment I due to the increased EXP diversity within a gallery. Otherwise, at least one of the two hypotheses is rejected.

**Results**. The PCA 3D plots for Dataset G1 were offered as supporting information. The plots support the desired purpose of the dataset for the experiments.

The DCNN model capacity to correctly classify Dataset G1 into galleries was discussed next. ANOVA post-hoc analysis exhibited four distinct behaviors ($p < 0.01$ or $p = 0$ in all tests). The box plot (in the supporting information) of the overall metrics along with their numerical values, e.g., MCC was 0.54, 0.30, 0.56 and 0.64, and ACC was 0.65, 0.46, 0.66 and 0.71, also confirmed the desired difficulty levels of the subsets. MCC values were 0.7 to 0.16 larger than ACC values.

The class-wise performance metrics in Fig 11 indicate a decreased DCNN performance as compared to the classification performance obtained for Dataset S1. PPV and TPR values were consistently low for all galleries, except Gallery "The Unknown", which offered the best

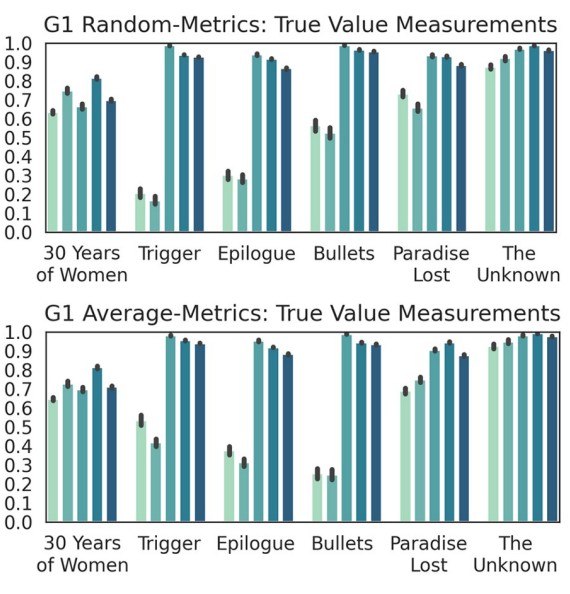

**Fig 11. Class-wise classification metrics results for dataset G1.**

performance for Dataset G1. Galleries "Trigger", "Epilogue", and "Bullets" were the hardest, second hardest, and third hardest to classify, as the experiment went from considering subset *random* to subset *easy*. Hence, the increased diversity of the within-gallery EXPs had an important influence on lowering the DCNN model's capacity to correctly identify a gallery. Actually, subset *difficult* was the hardest to classify among all subsets used in this work. The lower capacity of the DCNN model to correctly classify Dataset G1 support hypotheses I and II.

The detailed analysis of the results showed that for subset *difficult* of Gallery "The Unknown", PPV dropped while its TPR stayed high, suggesting that the number of FP increased. Images from other galleries were misclassified to this gallery. Also, the expectation for Gallery "30 Years of Women" was incorrect, as the DCNN model had an average performance for this gallery. One possible reason could be its large number of data points as compared to the other galleries. The expectation for Gallery "The Unknown" was correct despite its size being about 2.8 times smaller than that of Gallery "30 Years of Women". Future work will address the two unexpected situations.

## Experiment III. The impact of EXPs and NEXPs on distinguishing art images from non-art images

**Dataset S4 description**. This experiment investigated the validity of the two hypotheses when separating art images from non-art images, including the impact of the dataset sizes. As explained in the description of Experiment I, this experiment was added to explain the high performance obtained for Gallery "My Mother's Clothes" in Dataset S1. In addition to the galleries in Dataset S1, Dataset S4 included a new gallery of non-art images of ready-made, ordinary objects, like human clothes. These images were similar in their EXPs with galleries "Trigger" and "My Mother's Clothes", but had no NEXPs.

If hypotheses I and II were true then Dataset S4 would be harder to classify than Dataset S1, including having a lower performance for galleries "Trigger" and "My Mother's Clothes" than their classification performance obtained for Dataset S1. The performance obtained for Gallery "Non-Art" should be also low, as Gallery "Non-Art" has no NEXPs but presents similar EXPs like the two galleries above. Otherwise, at least one of the two hypotheses should be rejected.

To study the impact of the dataset sizes on the classification performance, experiments were run for two versions of Gallery "Non-Art", a 34-image version and an 18-image version. This experiment also addressed the expectation that the gallery size does not influence classification using NEXPs, but if two galleries have similar EXPs, the larger gallery is likely to offer better performance.

**Results**. The PCA 3D plots for Dataset S4 were given as supporting information. The plots reflect the desired purpose of the dataset.

The performance results for classifying Dataset S4 into galleries using the DCNN models were as follows. One-way ANOVA results for the two versions of Dataset S4 (e.g., with 34 and 18 extra non-art images) showed a statistically significant difference ($p < 0.01$). However, based on the box plot in Fig 12(b), the statistical sensitivity was negligible. The class-wise performance analysis of the two versions shows in Fig 12(a) identical trends aside from the expected bias due to Gallery "Non-Art". Therefore, the rest of the experiments focused only on the 34-image version of this gallery.

ANOVA post-hoc analysis showed no statistically significant difference between subsets *average* and *random* ($p_{Tukey\ HSD}$ = 0.001, 99% C.I = [0.0021, 0.0249]), $p_{Dunnett\ T3}$ = 0.002, 99% C.I = [0.0015, 0.0255]), and $p_{Games–Howell}$ = 0.002, 99% C.I = [0.0017, 0.0254]), and a strongly significant difference for the other subsets ($p < 0.01$ or $p = 0$ in all tests). The box plots (in the supporting information) for subsets *average* and *random* had small differences, like outliers

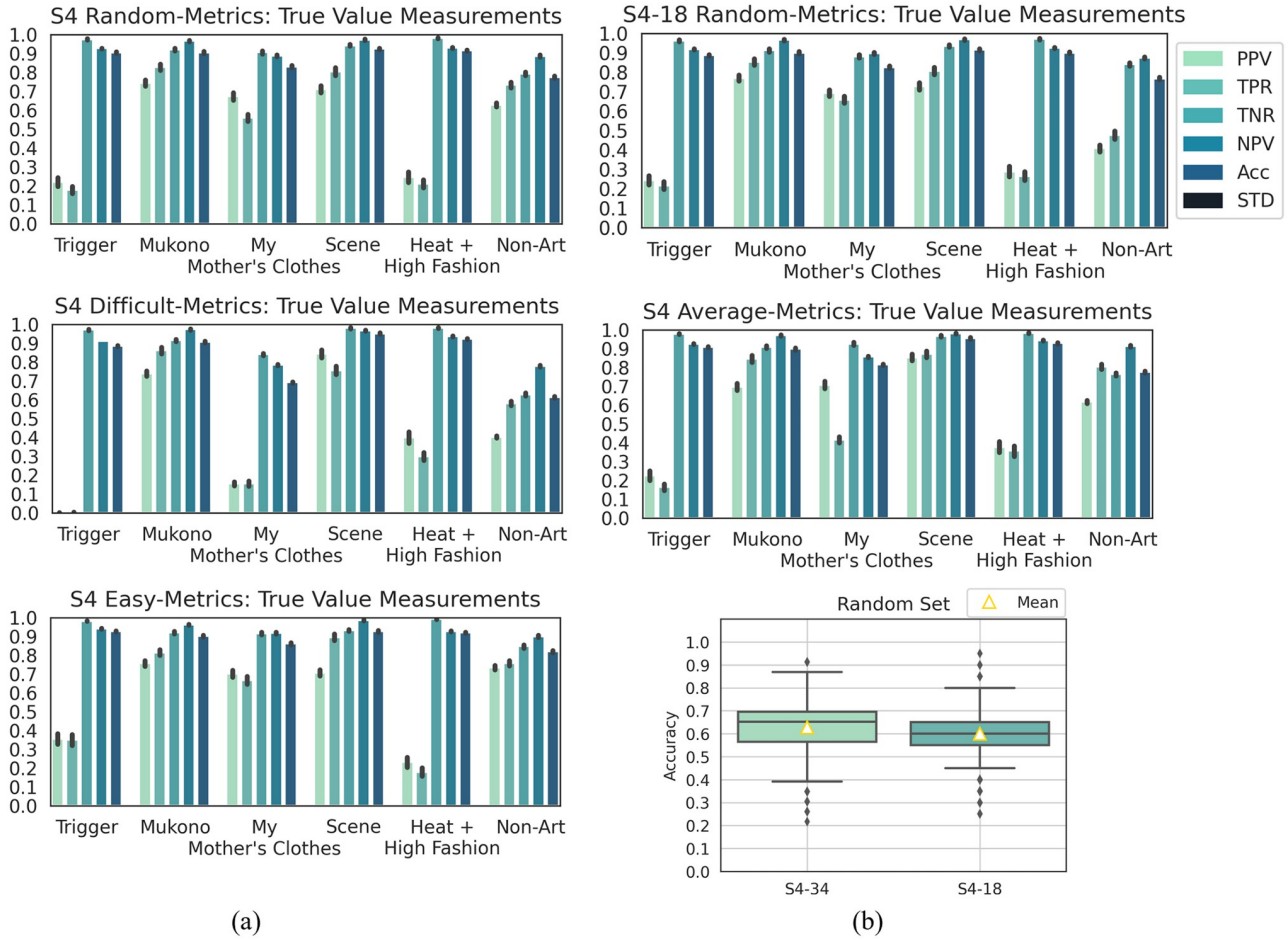

**Fig 12. Results for dataset S4 and "Non-Art"'s 34-image/18-image versions.** (a) Class-wise classification metrics results. (b) Box and whisker plots of the overall accuracies (ACC) of the DCNN classification for datasets S4–34 and S4–18.

and IQR, similar to the previous two experiments. Thus, the DCNN model had a distinct behavior for the four subsets.

The capacity of the DCNN model to correctly identify the galleries for the four subsets of Dataset S4 was discussed next. The class-wise metrics in Fig 12(a) indicated different distributions for the four subsets. Similar to set G1, MCC were 0.55, 0.35, 0.57 and 0.62, and ACC values were 0.63, 0.49, 0.64 and 0.68. The overall and class-wise performance confirmed the expectation that the non-art gallery confused the DCNN model. PPV and TPR values decreased for galleries "Trigger" and "My Mother's Clothes" as compared to their performance for Dataset S1. PPV and TPR values for Gallery "Non-Art" were low except subsets *random* and *easy* for which it was higher.

Specifically, the DCNN classification performance was the highest for galleries "Mukono" and "Scene" (for all their subsets), as they have distinct EXPs while they have little EXP similarity with Gallery "Non-Art". The larger sizes of galleries "Mukono" and "Scene" also explain why the model performed better for these galleries as compared to Gallery "Heat + High Fashion". Galleries "Trigger", "My Mother's Clothes", "Heat+ High Fashion", and "Non-Art" produced a low classification performance. The performance for subsets *easy* was better for all galleries, aside galleries "Trigger" and "Heat+ High Fashion", as these galleries contain

detectable EXPs that should differentiate art objects from non-art images of the same object. However, Experiment III showed that the DCNN model's learning of EXPs is insufficient.

## Discussion

Human experts assemble galleries and exhibitions based on their interpretations grounded in mental processing of the visual art images through their explicit and tacit knowledge obtained through formal training and experience, as well as the ideas specific to their context [54]. Some of their analysis and decisions can be explained through rules, like those summarized in art history [54], but other are subjective interpretations. It can be argued that there is currently no formalized, quantitatively-defined art ontology and procedural analysis method that could serve as the theoretical backbone for automatically understanding art, including artwork grouping into galleries based on its meaning, artist intention, and viewer interpretation. Instead, art galleries reflect a qualitative, narrative interpretation of art objects based on assembling EXPs into NEXPs that define the meaning, intention, and interpretation of artwork. This work suggests that general-purpose vision databases have likely only a limited role in curating art, such as to use them to train DNNs to recognize low-level features, because the meaning of art objects is absent (i.e. NEXPs).

Experiments showed that differentiating between difficulty levels (e.g., subsets *difficult*, *average*, and *easy*) is not cumulative, so that it can be easily quantified statistically, as there are no significant statistical differences between the subsets distinguished by our art expert.

While other research suggests that DCNNs can reliably learn object fragments and then use these fragments in art scene understanding [27], this work argues that learning does not include all object features needed to group related artwork into galleries. Features that define an object's uniqueness within an artwork are likely not learned, if they are not critical in recognizing the object from other objects. For example, a unique but repetitive combination of color on a grayscale image can be specific to an artist and help distinguish his work from other artwork. Due to its repetitive nature, a DCNN might learn the specific feature. However, rare features (e.g., EXPs) are not learned if they pertain to repetitive, high-level concepts (i.e NEXPs). Experiments showed that an artist's signature was not picked up by the DCNN model unless it was based on repetitive EXPs that could be learned, like having a yellow stripe over a greyscale image. A consequence of this observation is that aggregated, statistical metrics can observe global, systematic differences but not individual features. Histograms and outlier analysis, e.g., the number, position, and type of outliers, could address this limitation.

New metrics are required to capture the assessments of experts, like novelty, craftmanship, and viewer perception of artwork. These metrics must be conditioned by the cultural context of an expert's assessment.

Fig 13 summarizes the themes of some of the galleries used in the experiments. They include genres, like human figures and landscapes. Art objects having, but not necessarily, female figures as one of their central pieces addressed themes, like female identity, female fashion, African identity in the sixties and seventies, eighties, and contemporary. Fig 13 shows an ontology fragment of these concepts, in which arrows indicate the general—to specific relation and dashed lines the combination of concepts that co-occur in an art object. These relations are one kind of possible associated meanings, but other interpretations exist too. Extracting possible meanings for an art object includes identifying the symbolism of the concepts as well as conceptual interpretations, like analogies and metaphors, for the relations among concepts. Moreover, object EXPs, like color, shapes, texture, position, hues, illumination, and so on, can have a certain symbolism, interpretation, or induce a certain feeling to the viewer [54]. For example, common objects in an art composition could point to everyday life, and possibly to

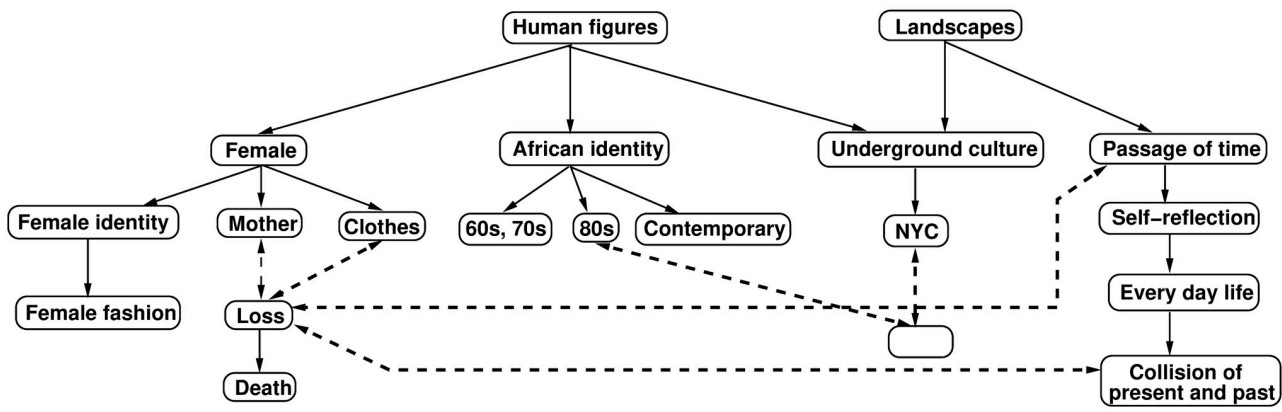

**Fig 13. Gallery themes summary.**

the collision of present and past [54]. Or, the relative positioning of objects or their unusual postures, e.g., a chair's position, can serve a certain purpose in an artwork's theme narrative [54]. Experiments suggest that DCNN classification difficulty relates to the ambiguity of how EXPs, i.e. the visuals of physical objects, relate to NEXPs and their higher-level semantics, like intention and interpretation, which is an artwork's projection into the idea space. The difficulty increases with the ontology's abstraction levels where ambiguities occur (Fig 13). Having multiple narratives for an object is also part of the possible ambiguities. The analysis of the differences between human art curation and DCNN classification shows several limitations of DCNN models in learning and understanding higher-level semantics. All learned differences are based on visual EXPs, like texture, tones, shapes, and objects. However, models are not capable to dynamically reprioritize the importance of EXPs depending on the process that would lead to understanding NEXPs and the meaning of an art object. For example, Baxandall explains that understanding art is a problem-solving process that constructs a narrative that expresses an object's meaning [54]. The object must be reinterpreted and reprioritized in the context of the narrative, while possibly dropping significant amounts of general-purpose learning using generic image databases, like ImageNet.

Another limitation of DCNNs related to NEXP learning refers to creating plausible narratives expressing the theme of an artwork. Narratives are based on the connections to the artist's or observer's context (including previous art), and the causal relationships between objects or their symbolic meaning, as well the mapping relations in the case of meanings based on analogies, metaphors, and abstractions [55, 56]. Some insight related to the historical context can be inferred using details, like clothing, hair style, or furniture. However, some of these details might not be captured during DCNN learning as they are less frequent than other features. Also, while recent methods can identify and learn some analogical mappings [23], these methods are symbolic and use numeric metrics to establish the mappings. Current DCNNs cannot learn well mappings, including some with qualitative, subjective, social, and emotional knowledge. A possibility would be to collect such data through surveys and then incorporate it into the DCNN learning process [12]. However, surveys are likely to be ineffective in helping to find which EXPs and NEXPs are the cause for the survey inputs, even though a human expert can indicate quite accurately how visual cues, like color and pattern, produce a certain interpretation or emotion.

Finally, there is similarity between art creation defined as open-ended problem solving and other creative processes, like engineering design. Problem framing in design relates to theme

selection in art, while creating the structure (architecture) of an engineering solution corresponds to creating the structure of a painting scene. The two solution spaces are constrained by various design rules and aesthetic rules, respectively, e.g., proportions, projections, coloring, and so on [54]. However, there are major differences too. While engineering is mostly guided by numerical performance values that express the objective quality of a design and to a much lesser degree by subjective factors, like preference for some functions, art creation is guided by arguably no quantitative analysis, being subjected only to qualitative, subjective evaluations. Besides, an engineering solution has a well-defined meaning and purpose, which is perceived in the same way by all. In contrast, the meaning of art depends on the artists and viewers, gets shaped by different cultures, and evolves over time.

## Conclusions

Modern theories of art suggest that Exhibited Properties (EXPs) and Non-Exhibited Properties (NEXPs) characterize any work of art. EXPs are visible features, like color, texture and form, and NEXPs are artistic aspects that result by relating an art object to human history, culture, the artist's intention, and the viewer's perception. Current work on using Deep Neural Network (DNN) models to computationally characterize artwork suggests that DNNs can learn EXPs and might gain some insight on meaning aspects tightly related to EXPs, but there are no extensive studies about the degree to which NEXPs are learned during DNN training, and then used for automated activities, like classifying artwork into galleries. To address this limitation, this work conducted a comprehensive set of experiments about the degree to which Deep Convolutional Neural Network (DCNN) models learn NEXPs of artwork. Two hypotheses were formulated to answer this question: The first hypotheses states that EXP similarities and differences within and between art galleries determine the difficulty level of DCNN classification. The second hypothesis states that DCNN models do not capture NEXPs well for art gallery classification.

Three experiments were devised and performed to verify the two hypotheses using datasets about art galleries assembled by an art expert. Experiments used the VGG-11 DCNN pretrained on ImageNet database, and then retrained using art images. The three experiments considered the following situations: (1) using EXPs and NEXPs for classification of art objects in solo (single artist) galleries, (2) utilizing EXPs and NEXPs for classification of art objects in group galleries, and (3) distinguishing art objects from non-art objects, and the impact of dataset size on classification results. Datasets were put together for each situation and for different difficulty levels of DCNN classification. Results were analyzed using statistical and classification measures.

The experimental study validated the two hypotheses. VGG-11 DCNN did not learn NEXPs sufficiently well to support accurate classification of modern artwork into galleries similar to those curated by human experts, and EXPs were insufficient for understanding, interpreting, and classifying artwork. Higher EXP similarity among galleries or higher EXP diversity within a gallery increased the difficulty level of classification in spite of their NEXP values, which suggests that EXPs were the determining factor in classification.

Dataset size was not a main factor in improving DCNN classification, but increasing dataset size can help galleries with similar EXPs.

This work suggests that any attempt to automate art understanding should be equipped with mechanisms to capture well both EXPs and NEXPs of artwork.

The three experimental studies are useful not only to characterize the general limitations of DCNN models, but also to understand if NEXPs of art objects can be distinguished only using their EXPs, thus if an art object is fully specified within its body of similar work, e.g., gallery, or

if NEXPs depend to a significant degree on elements not embodied into an art object, like contextual elements, the artist's intention, and the viewer's interpretation. Experimental results support the second perspective.

## Further research directions

The DCNN model studied in this work can arguably be a rough, qualitative predictor of artwork understanding by a person without artistic training. The model's art "knowledge" comes by superimposing features learned from a few art galleries on the features learned using images from the general vision domain. Experiments with DCNN models that would aggressively transfer knowledge from the art domain (and not only a few galleries) would add to the understanding of how well DCNNs can learn NEXPs. Another avenue of future work would consider other DNN models, such as VisionTransformer [56] and ConvNeXt [57], alongside with Transfer Learning techniques with a higher learning capacity, e.g., cascaded network architectures.

Finally, the design and analysis of the datasets and experiments could explore DCNN's preferences and biases, i.e. whether shape or color are more important in classification, or which features tend to be misclassified.

## Supporting information

**S1 Table. List of common terms in art history and visual arts to describe the EXPs.**
(PDF)

**S2 Table. Galleries' EXPs.**
(PDF)

**S3 Table. Galleries' NEXPs.**
(PDF)

**S4 Table. Dataset info: Galleries, images (number of images per gallery), total (total number of images per dataset).**
(PDF)

**S5 Table. Numerical values of statistical tests and measures.**
(PDF)

**S1 Fig. Galleries' outliers.**
(TIF)

**S2 Fig. Results for dataset S2.** (a) PCA plots for the training and test data of subsets difficult, average, and easy. (b) Box and whisker plots of the overall accuracies (ACC) of the DCNN classification of the four subsets.
(TIF)

**S3 Fig. Results for dataset S3.** (a) PCA plots for the test sets. (b) Box and whisker plots of the overall accuracies.
(TIF)

**S4 Fig. The results for dataset G1.** (a) PCA plots for test data for subsets difficult, average, and easy. (b) Box and whisker plot of overall accuracies (ACC) of the DCNN classification of the four subsets.
(TIF)

**S5 Fig. Results for dataset S4.** (a) The PCA plots for the training and test sets for subsets difficult, average, and easy. (b) The box and whisker plot of the overall accuracies (ACC) of the DCNN classification of the four subsets.
(TIF)

**S6 Fig. Box and whisker plot of overall accuracies for datasets SF1 to SF4.**
(TIF)

**S7 Fig. Class-wise metrics results for datasets SF1 to SF4.**
(TIF)

**S8 Fig. Accuracy histogram plots for all sets.** Histogram plots verify the distribution of the measured random variable.
(TIF)

**S9 Fig. Normal Q-Q plots for all sets.** Normal Q-Q plots are another method for verifying the distribution of the measured random variable.
(TIF)

**S10 Fig. Box plot summaries.** (a) Box and whisker plot of overall accuracies for all datasets random sets (b) Box and whisker plot of overall accuracies for all datasets handpicked sets (difficult, average, and easy).
(TIF)

**S11 Fig. Remaining class-wise metrics (FPR, FNR, and FDR).**
(TIF)

## Author Contributions

**Conceptualization:** Mahan Agha Zahedi, Niloofar Gholamrezaei, Alex Doboli.

**Data curation:** Mahan Agha Zahedi, Niloofar Gholamrezaei.

**Formal analysis:** Mahan Agha Zahedi, Niloofar Gholamrezaei, Alex Doboli.

**Investigation:** Mahan Agha Zahedi, Niloofar Gholamrezaei, Alex Doboli.

**Methodology:** Mahan Agha Zahedi, Niloofar Gholamrezaei, Alex Doboli.

**Project administration:** Mahan Agha Zahedi.

**Resources:** Mahan Agha Zahedi.

**Software:** Mahan Agha Zahedi.

**Supervision:** Mahan Agha Zahedi, Niloofar Gholamrezaei, Alex Doboli.

**Validation:** Mahan Agha Zahedi, Niloofar Gholamrezaei.

**Visualization:** Mahan Agha Zahedi, Niloofar Gholamrezaei.

**Writing – original draft:** Mahan Agha Zahedi, Niloofar Gholamrezaei, Alex Doboli.

**Writing – review & editing:** Mahan Agha Zahedi, Niloofar Gholamrezaei, Alex Doboli.

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
