## [Decision Letter · Decision Letter 0]

18 Mar 2024

PONE-D-23-31505How deep is your art: an experimental study on the limits of artistic understanding in a single-task, single-modality neural networkPLOS ONE

Dear Dr. Agha Zahedi,

Thank you for submitting your manuscript to PLOS ONE. After careful consideration, we feel that it has merit but does not fully meet PLOS ONE’s publication criteria as it currently stands. Therefore, we invite you to submit a revised version of the manuscript that addresses the points raised during the review process.

We look forward to receiving your revised manuscript.

Kind regards,

Dang N. H. Thanh, Ph.D

Academic Editor

PLOS ONE

Journal Requirements:

Did you know that depositing data in a repository is associated with up to a 25% citation advantage (https://doi.org/10.1371/journal.pone.0230416)? If you’ve not already done so, consider depositing your raw data in a repository to ensure your work is read, appreciated and cited by the largest possible audience. You’ll also earn an Accessible Data icon on your published paper if you deposit your data in any participating repository (https://plos.org/open-science/open-data/#accessible-data).

3. Please note that PLOS ONE has specific guidelines on code sharing for submissions in which author-generated code underpins the findings in the manuscript. In these cases, all author-generated code must be made available without restrictions upon publication of the work. 

Please review our guidelines at https://journals.plos.org/plosone/s/materials-and-software-sharing#loc-sharing-code and ensure that your code is shared in a way that follows best practice and facilitates reproducibility and reuse.

4. In the online submission form, you indicated that: "Datasets cannot be shared publicly because of copyright. However, we can provide the datasets upon request for academic/non profit purposes according to fair use act."

3. Uploaded as supplementary information.

5. Please amend the manuscript submission data (via Edit Submission) to include authors:

- Niloofar Gholamrezaei

- Alex Doboli

6. Please upload a copy of Figure 16, to which you refer in your text on page 23. If the figure is no longer to be included as part of the submission please remove all reference to it within the text.

7. We note that Figures and Supporting Figures in your submission contain copyrighted images:

- Fig_1

- Fig_4

- Table_4_Mukono

- Table_4_Trigger

- S2_Table_4_Bonsai

- S2_Table_4_Epilogue

- S2_Table_4_Familiar_Landscapes

- S2_Table_4_Hivernacle

- S2_Table_4_Little_Deaths

- S2_Table_4_Mukono

- S2_Table_4_Native

- S2_Table_4_Paradise_Lost

- S2_Table_4_Private

- S2_Table_4_The_Fall_of_Spring_Hill

- S2_Table_4_The_Fallen_Fawn

- S2_Table_4_Trigger

All PLOS content is published under the Creative Commons Attribution License (CC BY 4.0), which means that the manuscript, images, and Supporting Information files will be freely available online, and any third party is permitted to access, download, copy, distribute, and use these materials in any way, even commercially, with proper attribution. For more information, see our copyright guidelines: http://journals.plos.org/plosone/s/licenses-and-copyright.

(1) You may seek permission from the original copyright holder of those figures & supporting figures mentioned above to publish the content specifically under the CC BY 4.0 license. 

(2) If you are unable to obtain permission from the original copyright holder to publish these figures under the CC BY 4.0 license or if the copyright holder’s requirements are incompatible with the CC BY 4.0 license, please either i) remove the figure or ii) supply a replacement figure that complies with the CC BY 4.0 license. Please check copyright information on all replacement figures and update the figure caption with source information. 

If applicable, please specify in the figure caption text when a figure is similar but not identical to the original image and is therefore for illustrative purposes only.

8. We notice that your supplementary figures (Table_4_Mukono and Table_4_Trigger) are uploaded with the file type 'Figure'. Please remove the file and leave only the supplementary figures (S2_Table_4_Mukono and S2_Table_4_Trigger) uploaded. Please ensure that each Supporting Information file has a legend listed in the manuscript after the references list.

Reviewers' comments:

Reviewer's Responses to Questions

**Comments to the Author**

1. Is the manuscript technically sound, and do the data support the conclusions?

Reviewer #1: Yes

Reviewer #2: Yes

2. Has the statistical analysis been performed appropriately and rigorously? 

Reviewer #1: Yes

Reviewer #2: Yes

3. Have the authors made all data underlying the findings in their manuscript fully available?

Reviewer #1: Yes

Reviewer #2: Yes

4. Is the manuscript presented in an intelligible fashion and written in standard English?

Reviewer #1: Yes

Reviewer #2: No

5. Review Comments to the Author

Reviewer #1: The research article presents a thought-provoking investigation into the capabilities of Deep Convolutional Neural Networks (DCNN) in interpreting modern conceptual artwork. The study addresses the intricate nature of art interpretation, which is known for its multidimensional and subjective characteristics.

One of the key strengths of this paper is the clear articulation of two hypotheses regarding the classification of artwork properties by the DCNN model. The hypotheses suggest that the model utilizes Exhibited Properties, such as shape and color, for classification, while ignoring Non-Exhibited Properties like historical context and artist intention. Through a meticulously designed methodology, the experimental results supported these hypotheses, highlighting the DCNN's focus on exhibited properties for artwork classification.

However, the innovations in this paper focus on transitional convolution neural network, and the methods and theoretical innovations are not sufficient. The author must dig deeper into the innovation points before the article can be accepted by PONE.

Secondary, this paper offers valuable insights into the intersection of art and artificial intelligence, raising important questions about the boundaries of artistic understanding in neural networks. This paper should do more research in the area of intricate relationship between technology and creativity, for the sake of better explanation about the significance and necessity of your research

Thirdly, authors should present some picture about the art in your paper in order to better understand.

Finally, authors should share their code and project for easier reproducibility and persuasion.

Reviewer #2: The manuscript "How Deep is Your Art: An Experimental Study on the Limits of Artistic Understanding in a Single-Task, Single-Modality Neural Network" investigates the shortcomings in the Deep Convolutional Neural Network (DCNN) models by experimenting the degree to which a DCNN model can correctly distinguish modern conceptual artwork into the galleries taking into account Exhibited Properties for classification (shape and color) but not Non-Exhibited Properties (historical context and artist intention).

Based on previous comments, the manuscript has improved significantly; however, some typos and wrong sentence structure can be improved. Also, some improvement is needed in presenting the number of images in each of the datasets/galleries clearly and scaling them to fit the journal template guidelines. My other concern is whether the manuscript exceeds the page limits. Overall, the manuscript can be more structured and aligned to make the paper easy to follow.

6. PLOS authors have the option to publish the peer review history of their article (what does this mean?). If published, this will include your full peer review and any attached files.

Reviewer #1: No

Reviewer #2: No

---

## [Author Response · Author response to Decision Letter 0]

13 May 2024

The authors would like to thank the PLOS ONE editorial board. The response to the reviewer's comments were applied in the manuscript. A reviewer's response letter is submitted.

---

## [Decision Letter · Decision Letter 1]

10 Jun 2024

How deep is your art: an experimental study on the limits of artistic understanding in a single-task, single-modality neural network

PONE-D-23-31505R1

Dear Dr. Agha Zahedi,

We’re pleased to inform you that your manuscript has been judged scientifically suitable for publication and will be formally accepted for publication once it meets all outstanding technical requirements.

Kind regards,

Dang N. H. Thanh, Ph.D

Academic Editor

PLOS ONE

Additional Editor Comments (optional):

As recommendation from the reviewers and after checking the revision, I recommend acceptance of the paper.

Reviewers' comments:

Reviewer's Responses to Questions

**Comments to the Author**

1. If the authors have adequately addressed your comments raised in a previous round of review and you feel that this manuscript is now acceptable for publication, you may indicate that here to bypass the “Comments to the Author” section, enter your conflict of interest statement in the “Confidential to Editor” section, and submit your "Accept" recommendation.

Reviewer #1: All comments have been addressed

2. Is the manuscript technically sound, and do the data support the conclusions?

Reviewer #1: Yes

3. Has the statistical analysis been performed appropriately and rigorously? 

Reviewer #1: Yes

4. Have the authors made all data underlying the findings in their manuscript fully available?

Reviewer #1: Yes

5. Is the manuscript presented in an intelligible fashion and written in standard English?

Reviewer #1: Yes

6. Review Comments to the Author

Reviewer #1: The authors have answered and revised all my concerned question, so I have no more questions anymore in this step.

7. PLOS authors have the option to publish the peer review history of their article (what does this mean?). If published, this will include your full peer review and any attached files.

Reviewer #1: No
